# Corneal dysfunction precedes the onset of hyperglycemia in a mouse model of diet-induced obesity

**Aubrey Hargrave**[1¤]*, **Justin A. Courson**[1], **Vanna Pham**[1], **Paul Landry**[1], **Sri Magadi**[1], **Pooja Shankar**[1], **Sam Hanlon**[1], **Apoorva Das**[1], **Rolando E. Rumbaut**[2,3], **C. Wayne Smith**[3], **Alan R. Burns**[1,2]

**1** College of Optometry, University of Houston, Houston, Texas, United States of America, **2** Children's Nutrition Research Center, Baylor College of Medicine, Houston, Texas, United States of America, **3** Center for Translational Research on Inflammatory Diseases (CTRID), Michael E. DeBakey Veterans Affairs Medical Center, Houston, Texas, United States of America

¤ Current address: Department of Ophthalmology, Stanford University, Palo Alto, California, United States of America

* aubreych@stanford.edu

## Abstract

### Purpose

The purpose of this study was to use a mouse model of diet-induced obesity to determine if corneal dysfunction begins prior to the onset of sustained hyperglycemia and if the dysfunction is ameliorated by diet reversal.

### Methods

Six-week-old male C57BL/6 mice were fed a high fat diet (HFD) or a normal diet (ND) for 5–15 weeks. Diet reversal (DiR) mice were fed a HFD for 5 weeks, followed by a ND for 5 or 10 weeks. Corneal sensitivity was determined using aesthesiometry. Corneal cytokine expression was analyzed using a 32-plex Luminex assay. Excised corneas were prepared for immunofluorescence microscopy to evaluate diet-induced changes and wound healing. For wounding studies, mice were fed a HFD or a ND for 10 days prior to receiving a central 2mm corneal abrasion.

### Results

After 10 days of HFD consumption, corneal sensitivity declined. By 10 weeks, expression of corneal inflammatory mediators increased and nerve density declined. While diet reversal restored nerve density and sensitivity, the corneas remained in a heightened inflammatory state. After 10 days on the HFD, corneal circadian rhythms (limbal neutrophil accumulation, epithelial cell division and *Rev-erbα* expression) were blunted. Similarly, leukocyte recruitment after wounding was dysregulated and accompanied by delays in wound closure and nerve recovery.

**Data Availability Statement:** All relevant data are within the manuscript and its Supporting Information files.

**Funding:** This study was supported by NIH (https://www.nih.gov/grants-funding) and NEI (https://www.nei.nih.gov/grants-and-training) funding: NIH EY018239 (ARB, CWS and RER), P30EY007551 (ARB) and HL079368 (RER). Additional support from the U.S. Dept. of Veterans Affairs Merit Review Award I01 BX002551 (RER) (https://www.research.va.gov/services/shared_docs/resources.cfm#5). The funders had no role in study design, data collection and analysis, decision to publish, or preparation of the manuscript.

**Competing interests:** The authors have declared that no competing interests exist.

## Conclusion

In the mouse, obesogenic diet consumption results in corneal dysfunction that precedes the onset of sustained hyperglycemia. Diet reversal only partially ameliorated this dysfunction, suggesting a HFD diet may have a lasting negative impact on corneal health that is resistant to dietary therapeutic intervention.

## Introduction

Diabetic keratopathy is a carefully documented clinical problem in up to 70% of diabetics [1–4], affecting the safety of therapeutic interventions and the long-term health of the cornea [5–7]. The obesity epidemic has increased the incidence of type 2 diabetes worldwide [8], and is of particular concern because of marked increases in the early pediatric age range [9–11]. A recent CDC report [12] indicated approximately one in three U.S. adults over 19 years of age (estimated 79 million people) has prediabetes, a condition with blood glucose or hemoglobin A1c levels higher than normal but not high enough to be classed as diabetes. While research has focused heavily on the contributions of hyperglycemia and insulin resistance [13–21], it is clear that a high percentage of obese individuals exhibit a prediabetic metabolic syndrome with systemic inflammatory changes in adipose, muscle, liver and other tissues, contributing to diverse pathologies [22–28]. Major inflammatory changes include elevated proinflammatory cytokines in blood, a systemic Th17 bias in inflammatory responses [29–37], dysregulation of T cells (including γδ T cells) [38–40] and proinflammatory (M1) polarization of tissue macrophages [41–43]. Given the prevalence of obesity in the adult and pediatric populations [9, 11] and the knowledge that corneal healing after injury [44–48] or surgery [5, 7] is a complex and carefully regulated process, identifying potential therapeutic targets based on a better understanding of the pathogenesis of metabolic syndrome-associated keratopathy is important.

   Preclinical studies of type 2 diabetes in animals have confirmed and extended observations potentially relevant to human keratopathy [14–21, 49]. These models have been limited in considering the possibility of pathogenic early diet-induced changes that precede the development of type 2 diabetes. In the present study, we employed a model of diet-induced obesity in wild-type mice [22–28, 50, 51] to assess corneal dysfunction preceding the development of sustained hyperglycemia. Here we provide evidence of early corneal changes in homeostasis and wound healing after the consumption of an obesogenic diet.

## Materials and methods

### Mice

Six-week-old male C57BL/6 mice were fed an *ad libitum* (ad lib) high fat diet (HFD; 42% kcal milk-fat; Diet #112734; Dyets Inc., Bethlehem, PA) for 5, 10, or 15 weeks. Age-matched littermate mice on a normal chow diet (ND; 15% kcal fat; Advanced Protocol PicoLab Select Rodent 50 IF/6F 5V5R; LabDiet, St Louis, MO) served as controls. Diet reversal (DiR) mice were fed a HFD for 5 weeks followed by a ND for 5 weeks. For circadian studies, mice were fed a HFD or ND for 10 days. Our previous studies established that male mice exhibit a significant insulinemia after 24 weeks on the HFD without evidence of hyperglycemia [23]. Mice were housed in the Children's Nutrition Research Center vivarium in a temperature-controlled, 12h light-dark environment and were treated according to the guidelines described in the

Association for Research in Vision and Ophthalmology Statement for Use of Animals in Vision and Ophthalmic Research. The Animal Care and Use ethics committees at Baylor College of Medicine (IACUC #AN2721), and at the University of Houston (IACUC #16–005) approved this study.

## Body composition, metabolism and systemic inflammation analyses

Body composition was determined using a quantitative magnetic resonance (qMR) machine, Echo MRI-100 (EchoMRI, Houston, TX). Mice were weighed and their full body mass (g) was recorded immediately prior to euthanasia. Animals were euthanized by either $CO_2$ inhalation or isoflurane overdose, followed by cervical dislocation according to IACUC guidelines. Intra-abdominal epididymal adipose tissue (eAT) and liver were collected and weighed. For flow cytometry, adipose tissue was minced into pieces and digested with 280 U/ml collagenase type I (Worthington Biochemical, Lakewood, NJ, USA) in 1% BSA in PBS (phosphate buffered saline) for 75 min at 37˚C. Adipose macrophage subsets were identified using antibodies against: CD11c (HL3) from BD Biosciences, San Diego, CA, USA, F4/80 (BM8) and TNF-α (MP6-XT22) from eBioscience, San Diego, CA, USA and CD206 (MR5D3) from AbD Serotec, Kidlington, United Kingdom [52].

Blood leukocyte subsets were identified with antibodies against: Ly6G (1A8), CD41 (MWReg30) and CD3 (145-2c11) from BD Biosciences and F4/80 (BM8) from eBioscience. In some cases, after a 5h fast, blood was collected and the plasma levels of glucose and insulin were determined. Blood glucose levels were measured using a OneTouch Ultra glucometer (LifeScan, Milpitas, CA, USA) and plasma insulin levels were measured using a mouse insulin ELISA kit (EMD Millipore, Billerica, MA, USA).

## Mouse model for corneal wound healing

Mice were fed a HFD or ND for 10 days. As previously described, mice were anesthetized by intraperitoneal injection of pentobarbital sodium solution (40 mg/kg body weight; Nembutal, Ovation Pharmaceuticals, Deerfield, IL) before the central corneal abrasion was performed. The central corneal epithelium was marked with a 2-mm trephine and was abraded using a golf club spud (Accutome, Malvern, PA) under a dissecting microscope. For wound closure, fluorescein-stained injured corneas were imaged with a digital camera at time of injury and every 6h, starting 18h after injury until healed [52]. Fluorescein pools in the open wound, allowing the wound area to be measured using Fiji software [53]. The size of the wound at each time point was expressed as a percentage of the original wound area.

## Immunofluorescence imaging and deconvolution microscopy

Immunostaining was performed on whole mount corneas as previously described [54]. Briefly, corneas were dissected from whole eyes and radial cuts were made to flatten the cornea. Corneas were fixed with buffered 2% paraformaldehyde (EMS, Hatfield, PA) for 1h, washed 3 x 5 min with PBS, and then permeabilized and blocked with PBS containing 0.1% Triton X-100 and 2% bovine serum albumin for 30 min. Corneas were incubated with fluorescent-labeled antibodies overnight at 4˚C and then washed with PBS before being mounted in Airvol 205 (Sigma Aldrich, St. Louis, MO). Image analysis and quantification were performed using a DeltaVision microscope (Applied Precision, Carlsbad, CA). Corneal epithelial nuclei, and those nuclei showing a mitotic figure, were visualized using DAPI (Sigma Aldrich, St. Louis, MO); corneal nerves and NK cells were stained with anti-beta tubulin III (Tuj-1) and anti-Nkp46, respectively (R&D Systems, Inc., Minneapolis, MN); blood vessels, platelets and γδ T cells were stained with anti-CD31 (MEC13.3), anti-CD41 (MWReg30) and anti-γδ T cell

receptor (GL3), respectively (BD Biosciences, San Jose, CA); and neutrophils with anti-LY6G (Invitrogen, Carlsbad, CA). Limbal venule diameter measurements and counts of platelets, neutrophils, γδ T cells, NK cells and dividing epithelial cells were made from images captured with a 20X objective.

### Nerve morphology and function

A central region (200 μm x 200 μm) of the cornea was identified using x- and y-coordinates based on the location of the limbus. Images for nerve analysis were acquired as digital image z-stacks (0.3-μm/slice) spanning the entire thickness of the corneal epithelium using a 30X silicon oil objective. The subbasal plexus and epithelial branches were divided into separate groups using anatomical cues, each defined by a series of deconvolved images that were projected into a single maximum intensity image. Nerve density was estimated from the projected images using a custom MATLAB program, which uses edge detection to calculate the total nerve axon length in each image [55]. The densities of the epithelial branches and subbasal plexus were determined separately and combined to give the total nerve density for each cornea. Vertical nerves (connecting the subbasal plexus with the epithelial branches) were manually counted in each image z-stack. Nerve function was measured by testing the corneal sensitivity to tactile sensation using a Cochet-Bonnet aesthesiometer (12/100 nylon thread; Richmond Products, Albuquerque, NM). The threshold pressure response $(g/mm^2)$ for the central cornea was obtained by starting at a pressure too low for detection and then systematically increasing the pressure (shortening the monofilament length) until a blink was observed. This pressure response was tested twice with two observers confirming the occurrence of a blink.

### Epithelial cross-sections

To measure epithelial thickness, corneas from ND and HFD mice were collected prior to injury and 96h after wounding. Corneas were fixed with 0.1M sodium cacodylate buffer (pH 7.2) containing 2.5% glutaraldehyde (EMS, Hatfield, PA), post-fixed with aqueous 1% osmium tetroxide, dehydrated through an acetone series, and embedded in Embed812 resin (Polysciences, Warrington, PA). Transverse resin sections through the middle of the cornea were stained with 1% toluidine blue O dye and examined by light microscopy with a 20X objective lens. Epithelial thickness was measured from the top of the corneal epithelium to the basement membrane. Three central measurements (separated laterally by 50 μm) were taken from each section and averaged to get a single central thickness value for each cornea.

### qPCR analysis of *Rev-erbα*

Corneas were collected from 10d ND and HFD mice. Corneas from six to eight mice were pooled for each time point, with two samples per time point. Samples were cut into small pieces and then flash frozen. Total RNA was isolated with RNeasy columns (Qiagen, Germantown, MD) and 500 ng RNA was reverse transcribed into cDNA with Moloney murine leukemia virus RT (Applied Biosystems, Foster City, CA) for qPCR using TaqMan probes (Applied Biosystems, ThermoFisher, Grand Island, NY). Gene expression levels were normalized to GAPDH.

### Protein extraction and Luminex multiplex assay

Corneas were collected from ND (10w), HFD (10w), and DiR (5w HFD + 5w ND) mice. Whole corneas were pooled (3–7 corneas with 1 cornea from each mouse per pooled sample)

and homogenized prior to analysis; protein concentration was normalized to 15 μg/ml for each sample. Additional corneas were collected after 10 weeks on the diet and incubated in RPMI 1640 medium containing HEPES & L-Glutamine (EMD Millipore, Temecula, CA) for 6h to evaluate secreted products. Corneas and supernatants were flash frozen using liquid nitrogen and stored at -80˚C until use. All samples were tested for 32 analytes using a multiplex Luminex assay (MCYTMAG-70K-PX32; EMD Millipore, Temecula, CA).

## Statistics

The data are summarized as means ± standard deviation (SD) or standard error of the mean (SEM) as noted. Data analyses were performed using a Pearson correlation, a two-tailed two-sample independent t-test, a one-way ANOVA, or a two-way ANOVA as needed. Multiple comparisons were analyzed using the Tukey or Dunnett post-test (one-way ANOVA), and the Sidak or Bonferroni post-tests (two-way ANOVA). For data sets with unequal variances, data were transformed (natural log) prior to analysis. All statistical testing was performed using an alpha level of 0.05. Calculations and analyses were performed using GraphPad Prism 6 software (GraphPad Software, Inc., La Jolla, CA).

## Results

### Systemic changes in body composition, metabolism and leukocytes

Systemic responses of C57BL/6 male mice given unrestricted access to the HFD used in the current study have been published [22, 23, 50, 51, 56]. Changes in the respiratory exchange ratio (RER) indicative of increased fat consumption are evident within one day on the diet, and the percentage of total body fat mass as determined by qMR is significantly increased within 3 days on the diet [57]. After 10 weeks on the HFD, the weight of the epididymal adipose tissue (eAT) of mice in the current study was significantly increased as expected (ND, 1.22 ± 0.17 g, n = 8; HFD, 2.47 ± 0.14 g, n = 5; p = 0.0003, Table 1). This increase in adiposity was accompanied not only by an increase in the total number of macrophages, but also an increase in macrophages expressing pro-inflammatory M1 markers (CD11c and TNF) [58] within the adipose tissue (fold increases in the following phenotypes: F4/80[+], 2.8, p = 0.007; F480[+] CD11c[+], 2.4, p = 0.024; and F480[+] TNF[+], 2.1, p = 0.014; n = 5–8 mice per group, Table 1). There was no change in the number of cells expressing the anti-inflammatory M2 marker (CD206, p = 0.75, Table 1). We previously showed at 10 and 24 weeks HFD feeding

**Table 1. Systemic effects of HFD and ND on C57BL/6 male mice after 10 weeks of feeding.**

|  | Normal Diet | High Fat Diet |
|---|---|---|
| **Body weight (g)** | 32.07 ± 0.83 | 41.68 ± 0.90[****] |
| **eAT weight (g)** | 1.22 ± 0.17 | 2.47 ± 0.14[***] |
| **F480+ cells** | 118197.75 ± 28379.81 | 324751.20 ± 65986.01[**] |
| **F480+/CD11c+ cells** | 15513.75 ± 3235.52 | 36822.00 ± 9067.47[*] |
| **F480+/TNF+ cells** | 80377.50 ± 19037.79 | 171910.80 ± 25538.78[*] |
| **F480+/CD206+ cells** | 2634.38 ± 404.67 | 2919.60 ± 899.14 |

Data are represented as means ± SEM.

[*]p<0.05

[**]p<0.01

[***]p<0.001

[****]p<0.0001.

that plasma insulin levels increase 3- and 8-fold, respectively, while fasting blood glucose levels remain unchanged at ~ 200 mg/dL or less [23, 51].

## Corneal changes in response to HFD feeding and diet reversal (DiR)

Analyses of central cornea innervation (Fig 1A–1E) in mice fed the ND revealed a small but significant age-related decline in nerve sensitivity (increased pressure required to elicit a blink when comparing 5 and 15 weeks feeding, p ≤0.05) and this was accompanied by a 48% decrease in the number of vertical nerves within the epithelium over the 15 week feeding period (Fig 1A & 1B). Mice on the HFD exhibited a significantly greater loss of sensitivity to touch, requiring a marked increase in pressure to induce a blink response (Fig 1A). Additionally, there was a greater reduction in vertical nerves in the central cornea after 5 and 10 weeks of HFD feeding when compared to ND controls (Fig 1B and 1E). Corneal nerve density (epithelial branches and subbasal plexus combined) also exhibited a significant reduction, but only after 10 weeks of HFD feeding compared to mice on the ND (Fig 1C).

To assess the reversibility of corneal changes in mice on the HFD, some mice were fed the diet for 5 weeks, a duration which induced significant changes in vertical nerve density and corneal sensitivity (Fig 1), then switched to the ND with continued feeding for an additional 5 weeks. The diet reversed (DiR) corneas were compared to corneas from mice fed a ND or HFD for 10 weeks (Fig 2A–2C). Diet reversal was protective in that nerve density and sensitivity in the corneas of DiR mice were similar to mice fed a ND for 10 weeks. Regardless of diet, corneal sensitivity and nerve density (number of vertical nerves and total length of the epithelial branches and subbasal plexus) correlated negatively with body weight at the end of the 10 week feeding (Fig 2D–2F). The pressure necessary to induce a blink response increased with increasing body weight, while nerve density (number of vertical nerves and total nerve length) decreased with increasing body weight.

To determine if a 10 week HFD feeding is associated with increased corneal inflammatory mediator expression, excised corneas were probed for 32 inflammatory mediators using a

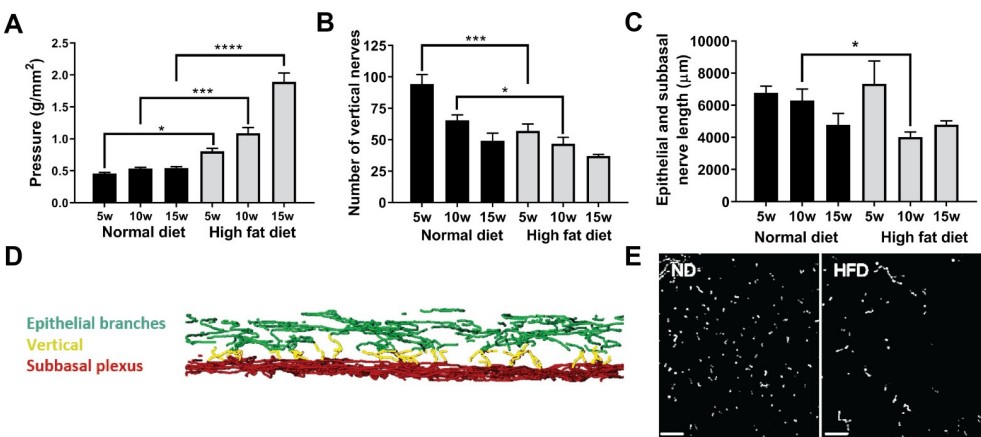

**Fig 1. Changes in corneal nerves and sensitivity with HFD.** C57BL/6 male mice were fed the ND or HFD for up to 15 weeks. (A) Cochet-Bonnett aesthesiometry was used to assess corneal sensitivity to touch before corneas were collected at 5, 10 or 15 weeks of feeding for corneal nerve analysis. Sensitivity is shown as the pressure necessary to induce a blink response (g/mm$^2$) (n = 28 ND, n = 35 HFD), (B) Morphometric analysis of the number of vertical nerves was determined by examining full thickness z-stacks of the epithelium (n = 12 ND, n = 13 HFD), (C) Analysis of the total length (μm) of epithelial nerves (epithelial branches and subbasal plexus) (n = 10 ND, n = 11 HFD), (D) Three-dimensional reconstruction of the epithelial nerves showing the subbasal plexus (red), epithelial nerve branches (green) and the vertical nerves (yellow) extending between them, and (E) *en face* projected image of the region of the epithelium containing vertical nerves (anti-β tubulin III, scale bar = 30 μm). Data shown as means ± SEM. *p<0.05, ***p<0.001, ****p<0.0001.

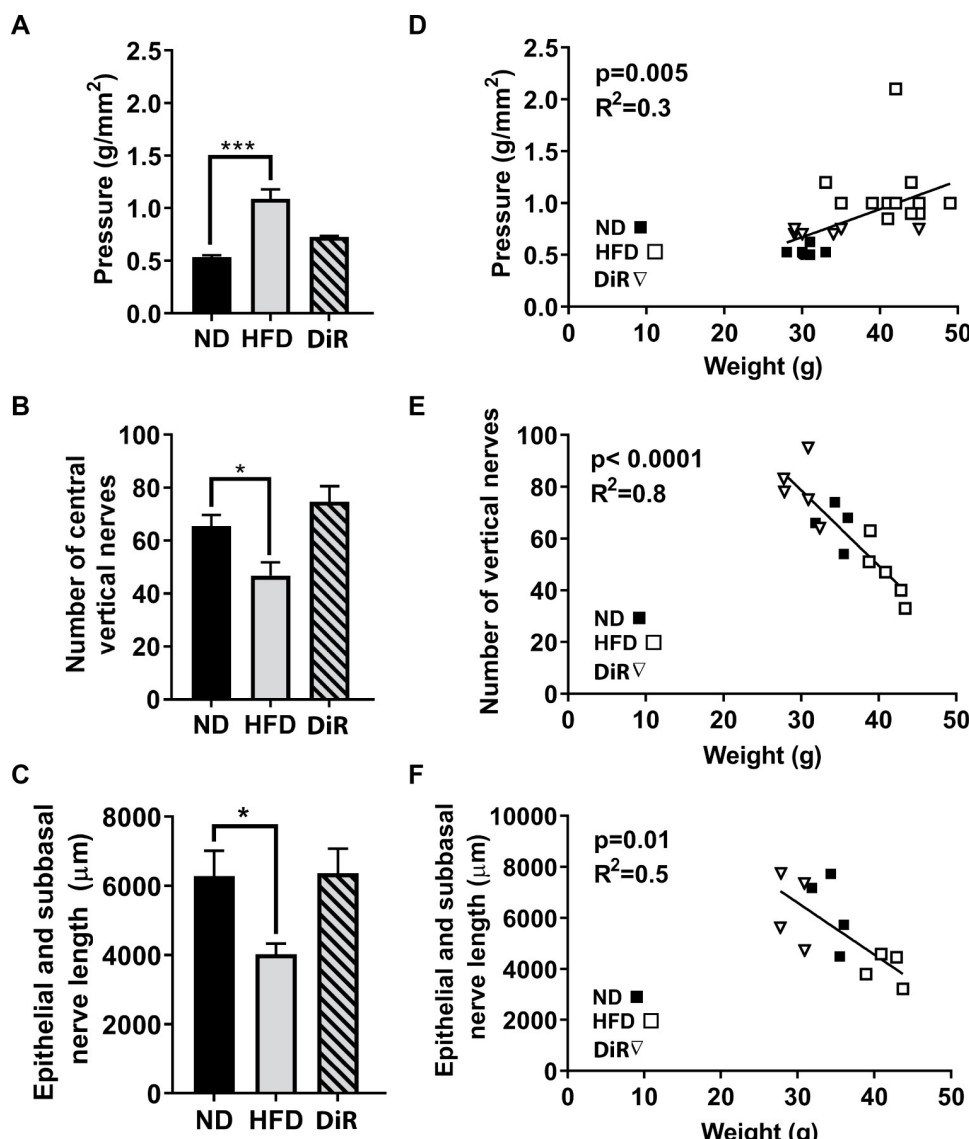

**Fig 2. Changes in corneal nerves and sensitivity with diet reversal.** C57BL/6 male mice were fed the ND or HFD for 10 weeks, and diet reversal (DiR) mice were fed the HFD for 5 weeks followed by the ND for 5 weeks. (A) Cochet-Bonnett aesthesiometry was used to assess corneal sensitivity to touch (g/mm$^2$) (n = 10 ND, n = 13 HFD, n = 12 DiR). Corneas were analyzed for corneal nerve density, including (B) number of vertical nerves (n = 4 ND, n = 5 HFD, n = 6 DiR) or (C) total length in μm of epithelial branches and the subbasal plexus (n = 4 per group). Data shown as means ± SEM. $^*$p<0.05, $^{***}$p<0.001. Pearson correlations of these measures with body weight for ND, HFD and DiR at 10 weeks are also plotted (D-F).

Luminex assay. In corneas analyzed immediately after excision (i.e., not cultured for 6h), three inflammatory mediators (IL-1α, IL-10 and IL-12p40) were elevated in mice fed a HFD compared to mice fed a ND (each p<0.05). By comparison, corneas from DiR mice were not different from those of ND, suggesting that the inflammatory state of the cornea had resolved. However, when the excised corneas from DiR mice were cultured for 6h, they showed significant increases in inflammatory mediator expression similar to the elevated levels found in cultured corneas from HFD mice (Fig 3A). Supernatants from the HFD and DiR corneas also revealed elevations in secreted mediators (Fig 3B). The sustained presence of inflammation in

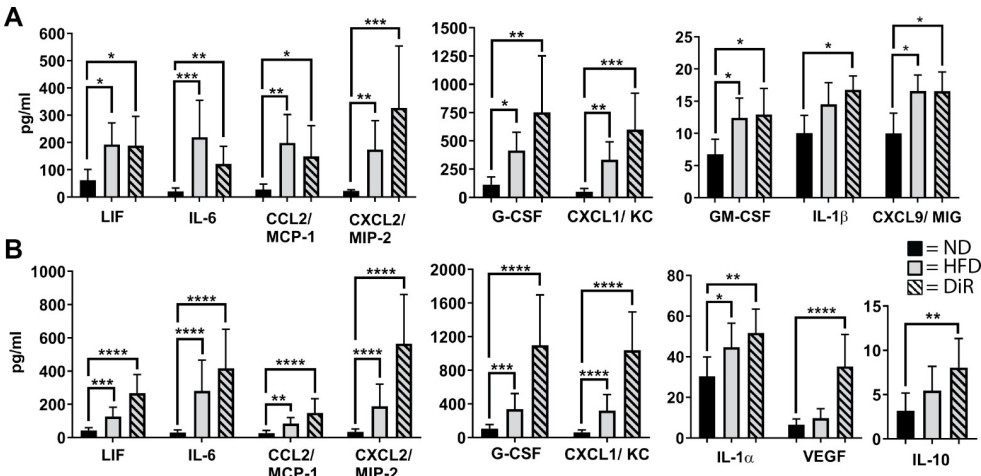

**Fig 3. Inflammatory protein expression in corneas.** C57BL/6 male mice were fed the ND or HFD for 10 weeks, and diet reversal (DiR) mice were fed the HFD for 5 weeks followed by the ND for 5 weeks. Corneas (n = 6 per group) were excised and cultured for 6h *in vitro*, and the (A) corneal extracts and (B) culture supernatant were analyzed for chemokines, cytokines and growth factors. Data shown as means ± SD. *p<0.05, **p<0.01, ***p<0.001, ****p<0.0001.

the DiR corneas was in contrast to the protective effects of DiR on weight gain and corneal nerve changes described above (Fig 2).

## Early corneal changes in response to HFD feeding for 10 days

To evaluate early HFD induced changes in the cornea, mice were studied on the tenth day after beginning the diet. The mice had significant increases in body weight and a 2-fold increase in the intra-abdominal epididymal fat pad weight and in body fat mass as measured by qMR (Table 2). No increase in fasting blood glucose was found in mice on the 10 day HFD, though plasma insulin was increased by 19% (p = 0.03). Flow cytometric evaluation of specific leukocyte subsets in the blood revealed a 2-fold increase in circulating numbers of lymphocytes and platelets (CD3+, p = 0.0033; CD41+, p = 0.047; n = 9 per group), but no significant changes in circulating numbers of monocytes or neutrophils (p>0.05 for $Ly6G^+$ and $F4/80^+$; n = 9 per group).

In ND mouse corneas collected at 10 days, extravascular neutrophils increased in the limbal stroma during the dark phase of the 24h cycle and then returned to low levels during the day

**Table 2. Effects of HFD and ND on C57BL/6 male mice after 10 days of feeding.**

|  | Normal Diet | High Fat Diet |
|---|---|---|
| Body weight (g) | 25.83 ± 0.34 | 27.67 ± 0.43 **** |
| eAT weight (g) | 0.39 ± 0.02 | 0.79 ± 0.08 **** |
| Fat mass (g) | 2.40 ± 0.13 | 4.90 ± 0.28 **** |
| Lean mass (g) | 23.38 ± 0.35 | 22.71 ± 0.30 |
| Liver weight (g) | 1.58 ± 0.04 | 1.51 ± 0.05 |
| Fasting Glucose (mg/dL) | 191.86 ± 16.59 | 208.47 ± 9.79 |
| Fasting Insulin (μg/ml) | 0.82 ± 0.06 | 1.01 ± 0.05 * |

Data represented as means ± SEM.

*p<0.05

****p<0.0001, n = 23 ND, n = 26 HFD.

(Fig 4A–4C). This influx of neutrophils failed to occur in mice on the HFD (Fig 4A & 4B). Dividing basal epithelial cells (i.e., cells undergoing mitosis) exhibited a reverse cycle from the neutrophil influx, with the highest level occurring during daylight hours (ZT 0–11) and the lowest level of mitosis occurring during the dark phase of the daily cycle (ZT 12–23) (Fig 4D). Mice on the HFD had significantly reduced numbers of mitotic cells during the daylight and dark phase hours. Peak epithelial division in mice on the ND occurred 5 hours earlier than in mice on the HFD; the nadir was delayed by 2 hours in HFD mice compared with those on a ND (Fig 4E). Additionally, the peak number of mitotic cells was approximately 25% less with the HFD feeding compared to the ND feeding. The normal circadian cycle of *Rev-erbα* expression in the cornea was suppressed in mice on the HFD (Fig 4F). While corneal sensitivity to touch did not show a circadian dependence, sensitivity was significantly reduced (increased pressure required to elicit a blink) in mice after 10 days on the HFD, even though epithelial nerve density (epithelial branches and subbasal plexus) was not reduced (Fig 4G).

## Influence of HFD feeding on epithelial healing after abrasion

Efficient wound closure in this experimental model is influenced by an inflammatory cascade involving chemokines, cytokines and the coordinated influx of T cell subsets, neutrophils, platelets and NK cells [22–28, 52]. In the current model, the response to central corneal epithelial abrasion was significantly altered after 10 days of HFD feeding. Limbal venule engorgement (increased diameter measured at 24h post-epithelial abrasion) did not occur when mice were fed the HFD (Fig 5A). In addition, epithelial cell division (mitosis) between 18 and 30h after corneal abrasion was significantly decreased (Fig 5B) and epithelial wound closure, heavily dependent on epithelial migration [59], showed a small, but significant delay from 18 to 24h post-injury (Fig 5C). Epithelial re-stratification and full-thickness recovery in this strain of mice usually occurs by 96h [59]. As expected, epithelial thickness at 96h post-injury was fully restored in mice fed a ND, whereas thickness recovery in mice fed a HFD was incomplete (p<0.01, Fig 5D).

Previous studies have shown γδ T cells to be important for the influx of neutrophils and platelets following corneal abrasion [60]. The γδ T cells increase in the stroma and epithelium, peaking within 12-18h after abrasion, and migrate in the epithelium toward the wound [60]. Influx of γδ T cells into the epithelium was evaluated in mice without wounding and at 12, 18 and 24h after central epithelial abrasion. The number of γδ T cells across the uninjured normal cornea is very low and was not different between mice fed a ND and a HFD. However, the influx of these cells into the epithelium was significantly blunted in mice on the HFD at each time point examined after corneal abrasion (Fig 6A).

Platelet extravasation at 12, 18 and 24h post-abrasion was significantly diminished at the corneal limbus of mice fed a HFD when compared to injured mice fed a ND (Fig 6B & 6C). Neutrophil migration into the central corneal stroma was evident at 12 to 30h post-abrasion in mice fed the ND, while mice on the HFD had significantly fewer neutrophils (Fig 6D). This is in marked contrast to the accumulation of neutrophils at the limbus, where significantly more limbal neutrophils accumulated in the mice fed the HFD between 6 and 96h post-abrasion compared to ND mice (Fig 6E & 6F). NK cells can modulate the inflammatory response and potentially limit neutrophil influx into the injured cornea [61], and mice fed a HFD had a slight but significant increase in limbal NK cell recruitment at 18 and 24h post-abrasion compared to similarly injured mice fed a ND (Fig 6G).

While the 10 day HFD reduced nerve sensitivity in unwounded corneas without measurable loss of nerve density in the epithelium (Fig 4G), recovery of nerve density and restoration of corneal sensitivity were both blunted in mice on the HFD at 96h following an epithelial abrasion (Fig 7A & 7B).

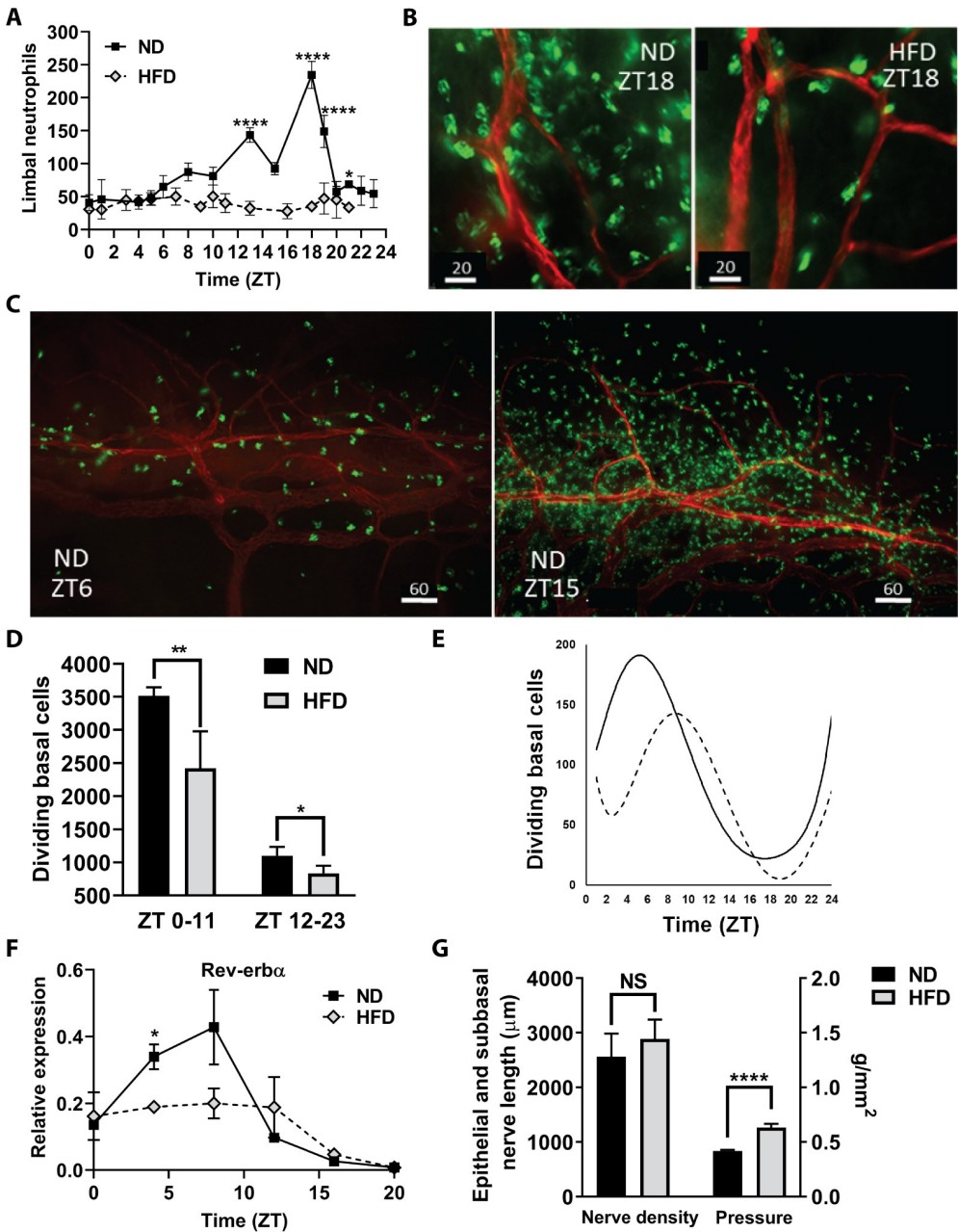

**Fig 4. Corneal changes after 10 days on the high fat diet.** (A) Neutrophils were counted in four 40X microscope fields of view in extravascular limbal tissue at intervals over a 24h cycle (n = 4 corneas per group at each time point). (B) Representative micrographs of neutrophils in the limbus at ZT18, comparing limbal regions of mice fed a ND and a HFD; neutrophils (green), limbal blood vessels (red), scale bar = 20 μm. (C) Representative micrographs of neutrophils around the limbal vascular network at ZT6 and ZT15 in corneas from mice on ND; neutrophils (green), limbal blood vessels (red), scale bar = 60 μm. (D) The average of the sum of mitotic cells observed in the basal epithelium in nine perpendicular (horizontal and vertical) 40X microscopic fields across the cornea from limbus to limbus between ZT0-ZT11 and ZT12-ZT23 (n = 4 corneas per group collected every hour for 24h). (E) Changes in the number of mitotic cells in corneal epithelial cells over a 24h cycle. A polynomial function was used to fit the curve for the dividing cell number (the average of the sum of mitotic cells counted in nine vertical and horizontal 40X fields at each time point) with n = 4 corneas collected every hour for 24h (ZT scale). Solid line = ND, dotted line = HFD. (F) Expression of clock gene *Rev-erbα* over a 24h cycle (n = 2 independent pooled samples from 6–8 mice at each time point). (G) Nerve density (total axon length (μm) of the epithelial branches and subbasal plexus) in the corneal epithelium of ND mice and HFD mice (n = 4 per group). Corneal sensitivity, measured as the tactile pressure (g/mm$^2$) required to elicit a blink, in mice fed the HFD for 10 days compared to ND fed mice (n = 6 per group). Data shown as means ± SD (A-E) and SEM (F, G). *p<0.05, **p<0.01, ****p<0.0001.

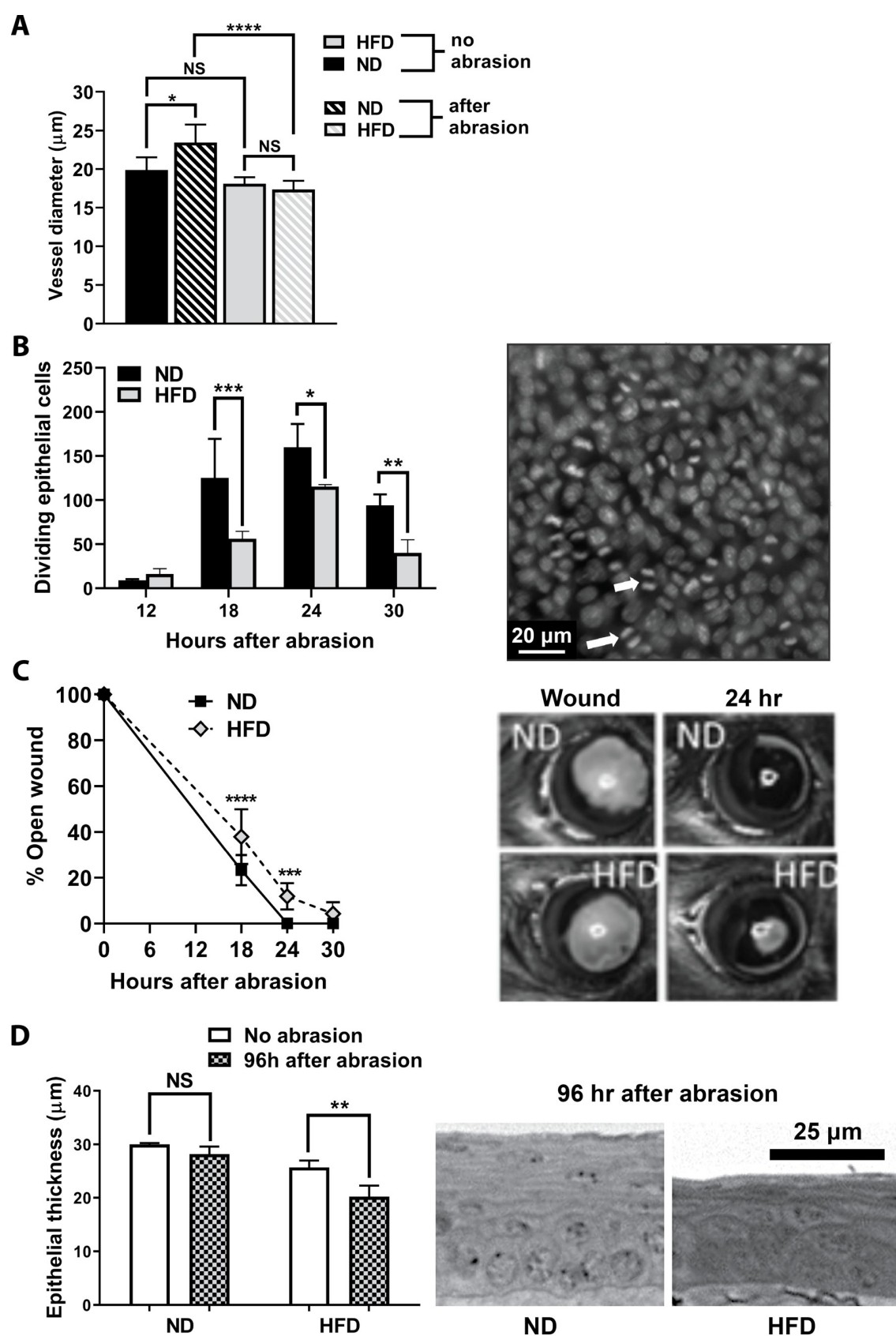

**Fig 5. Inflammatory response in the cornea after epithelial abrasion in ND and HFD fed mice.** (A) Venule diameters in the corneal limbus of mice fed a ND or a HFD before and 24h after receiving a corneal abrasion (n = 5 per group). (B) Dividing basal epithelial cells counted in nine perpendicular 40X fields of view across the cornea (n = 4 per group). Representative image of epithelial cell division in the basal epithelium at 18h after wounding (scale bar = 20 μm). Arrows indicate two examples of mitosis. (C) Wound closure was assessed by fluorescein staining of the open wound at different times after abrasion (n = 8 per group). Representative image of the epithelial wound immediately following abrasion (left) and 24h after abrasion (right) in ND and HFD mice. (D) Epithelial thickness (μm) was determined at 96h after epithelial abrasion in fixed and sectioned plastic embedded corneas (n = 4–5 per group). Representative images shown in the cross sections (scale bar = 25 μm). Data represented as means ± SD (A-C) and SEM (D). $^*p<0.05$, $^{**}p<0.01$, $^{***}p<0.001$, $^{****}p<0.0001$.

## Discussion

The purpose of this study was to assess corneal dysfunction in an experimental model of diet-induced obesity that precedes the development of sustained hyperglycemia. We showed changes in morphology and function after long-term HFD feeding for 5–15 weeks, as well as changes in circadian rhythm, corneal sensitivity, and the inflammatory response after 10 days of HFD feeding. There is recovery of nerve structure and function following diet reversal, but continued changes in inflammatory mediators suggest sustained dysfunction in the cornea.

Though mice on a ND show natural age-related declines in corneal nerve density [54] and corneal sensitivity, we found the mice fed a HFD for 5 to 15 weeks had an early and significant loss of corneal sensitivity and nerve density. These enhanced age-related declines in nerve density inversely correlate with body weight and adiposity. It appears that weight gain may be a predictor of increased corneal pathology during HFD consumption. Yorek et al. show a reduction in total corneal nerve fibers after 12 weeks of a HFD (60% kcal fat, lard-based), a time course similar to that used in the present study [62]. Additionally, aging and increasing adiposity are associated with increases in systemic inflammation [63, 64]. Expression of Growth Hormone Secretagogue Receptor (GHS-R), a receptor for Grehlin, consequently increases in adipose tissue during aging [63]. Ghrelin signaling is involved in M1 macrophage polarization, resulting in more inflammation [63]. Our finding that the cornea has a heightened propensity toward inflammation after short-term HFD (42% kcal fat, milk fat-based) feeding is consistent with these studies.

Monitoring corneal subbasal nerve density with an *in vivo* confocal microscope (IVCM) is considered an important tool for early detection of peripheral neuropathy in diabetic patients, given it is minimally invasive compared to traditional skin biopsies [65–72]. Our results indicate that nerve function is likely an earlier parameter of decline in mice on the HFD. The present study shows that a decline in sensitivity to tactile sensation occurs before detectable changes in subbasal nerve plexus morphology. Should our findings in the mouse translate to humans, monitoring corneal sensitivity in the clinic would be a minimally invasive technique to identify obese prediabetic patients with metabolic syndrome who are at risk for neurotrophic keratopathy.

Though we observed a measurable difference in nerve function, tactile sensation does not activate all nerve types in the cornea. The cornea is innervated by three distinct classes of peripheral sensory receptor neurons: mechanosensory, which respond only to mechanical forces; polymodal nociceptors, which respond to multiple stimuli including mechanical, heat, and chemicals; and cold thermoreceptor neurons, which respond to cooling and hyperosmolar solutions [73]. Approximately 51% of the corneal nerves in C57BL/6 mice are thought to respond to mechanical stimuli [73]. Alamri et al. showed that a 21% fat diet after 35–40 weeks has an effect on cold thermoreceptor nerve fiber density but does not affect polymodal nerve density [74]. Their study documents nerve loss attributable to fat in the diet. While the present study appears to support this conclusion, it is important to note that the two studies are not

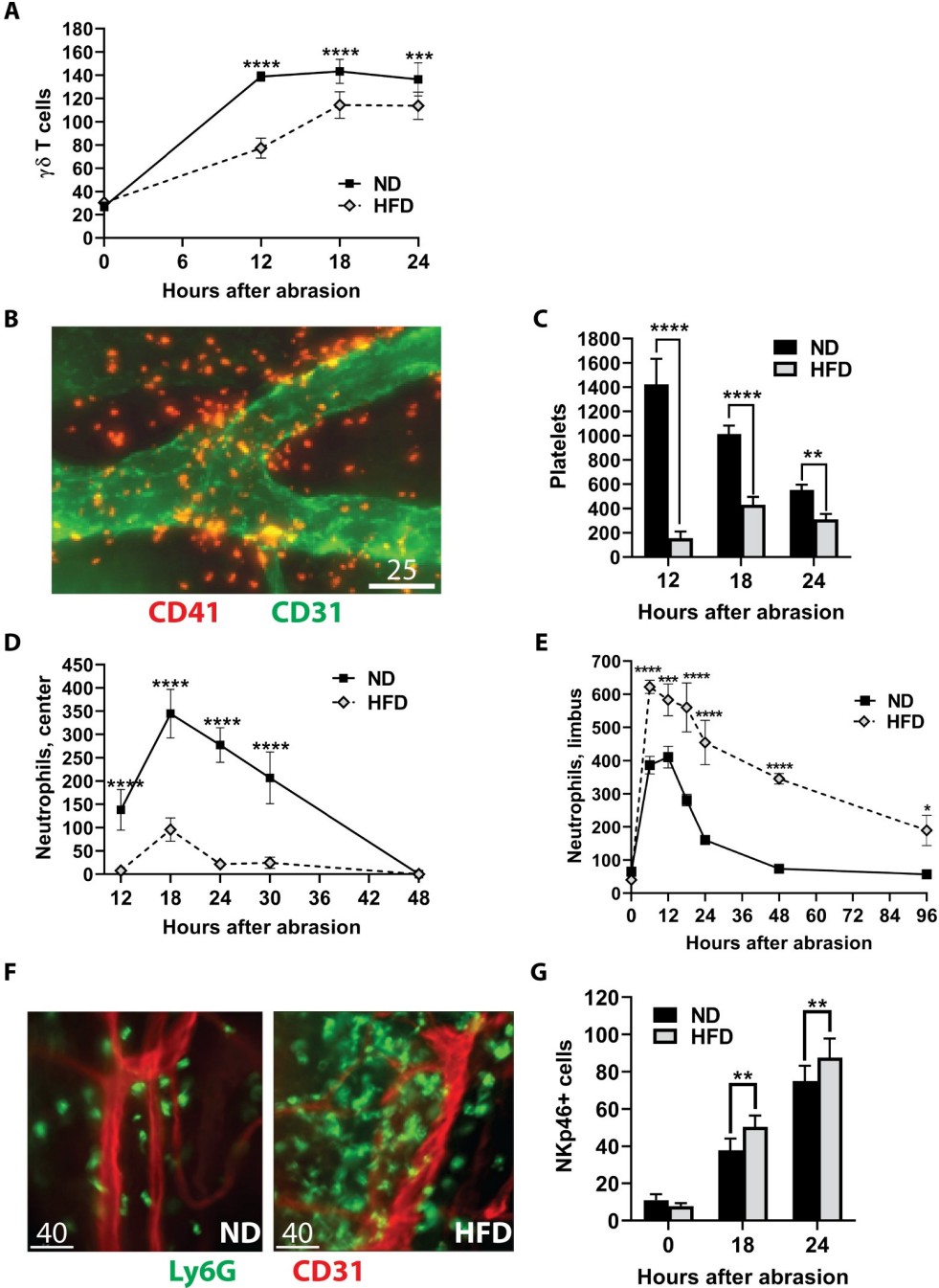

**Fig 6. Effect of the HFD on corneal responses to central epithelial abrasion.** Corneas from ND and HFD fed mice were analyzed after epithelial abrasion. (A) γδ T cells were counted in the epithelium and stroma across the cornea and plotted at 0, 6, 12, and 18h after epithelial abrasion (n = 6 per group). (B) Representative micrograph of extravascular platelets (red) around the limbal vascular network (green) at 12h after injury in a mouse fed a ND. Scale bar = 25 μm. (C) The sum of platelets in eight random non-overlapping 40X fields of view in the corneal limbus determined at 12, 18 and 24h after corneal abrasion (n = 4 at each time point). (D) Kinetics of neutrophil influx into the central abraded area of the cornea (counts from four 40X fields in the center of the cornea, n = 8 at each time point) are plotted at 12, 18, 24, 30, and 48h after abrasion. (E) Kinetics of neutrophil influx into the corneal limbus (counts from eight 40X fields in the limbus, n = 4–8 at each time point) prior to wounding (0h) and at 6, 12, 18, 24, 48, and 96h after abrasion. (F) Representative micrographs of neutrophil (green) influx around the limbal vascular (red) network at 48h after injury. Scale bar = 40 μm. (G) Limbal stromal NK cell counts (determined from nine vertical and horizontal 40X fields across the cornea, n = 8 per group) without wounding (0h), and at 18 and 24h after epithelial abrasion. Data shown as means ± SD. *p<0.05, **p<0.01, ***p<0.001, ****p<0.0001.

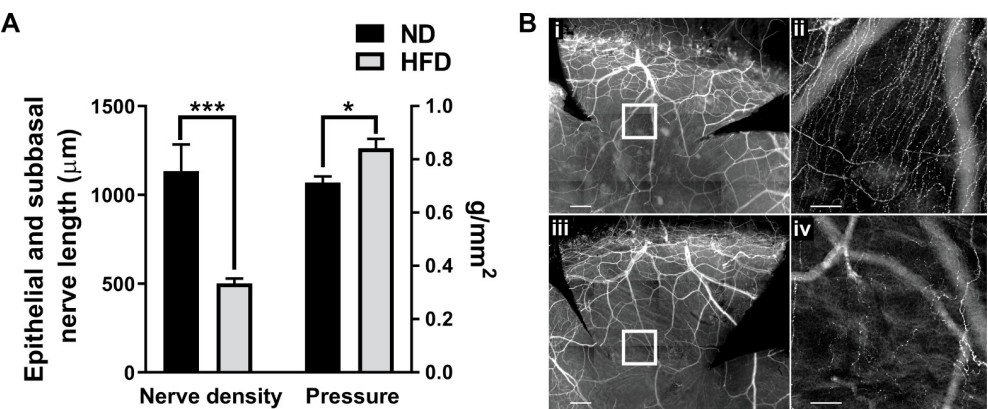

**Fig 7. Healing nerves after corneal epithelial abrasion.** Corneas from ND and HFD fed mice were analyzed after epithelial abrasion. (A) At 96h post-injury, nerve density (total axon length (μm) of the epithelial branches and subbasal plexus) recovery (left) in 10d HFD fed mice compared with those on the ND (n = 5 per group). Cochet-Bonnett aesthesiometry was used to assess corneal sensitivity to touch (g/mm²) at 96h post-injury (right) in 10d HFD mice compared to ND mice (n = 6 per group). (B) Corneal whole mounts showing corneal nerves of ND (i, ii) and HFD (iii, iv) mice at 96h post-injury. Corneas were stained with anti-β-tubulin III and imaged at 10X (i, iii; scale bar = 200 μm) or 20X (ii, iv; scale bar = 50 μm) magnification. The center of each inset box (i, iii; shown enlarged) is 500 μm from the center of the cornea (ii, iv) because regenerating nerves have not yet reached the center in any of the mice. Data shown as means ± SEM. *p<0.05, ***p<0.001.

directly comparable. The diet and time course differ (42% kcal from milk fat and low cholesterol for 10 days to 15 weeks in the present study compared to 40% kcal from clarified butter fat and high cholesterol for 35–40 weeks in the Alamri study). In addition, changes in mechanonociceptor neurons measured by aesthesiometry were not evaluated by Alamri and colleagues.

We observed elevated levels of IL-6 and MCP-1 in the cornea, consistent with the recognition of obesity as a low-grade systemic proinflammatory state. In obese adults and children, as well as animal models, this results from the secretion of cytokines, chemokines, and adipokines in adipose tissue, appearing as elevations in serum [50, 75]. In adipose tissues, it is the resident cells that respond to a HFD and produce cytokines and adipokines [76]. Following diet reversal from a HFD to a ND, we found that mouse corneas continued to show a heightened production of inflammatory mediators comparable to HFD corneas when they are excised and cultured for 6h. These results suggest that although there is improvement, continued corneal dysfunction is measurable when the microenvironment is perturbed (e.g., corneal excision and culture). Liu et al. reported pro-inflammatory (CCR2+) and anti-inflammatory (CCR2-) subsets of macrophages in the corneal limbus that contribute to wound healing and innate immunity [77]. While it remains to be determined if HFD feeding causes a change in limbal macrophage secretion of cytokines and chemokines, macrophages are likely candidates based on studies of other tissues where they are among the primary cells producing cytokines and chemokines, exacerbating systemic inflammation by switching to a pro-inflammatory (M1) phenotype [76, 78, 79].

HFD affects the cornea within 10 days of feeding, preventing the circadian influx of neutrophils into the limbal stroma, reducing basal epithelial cell division, reducing the daily cycle of *Rev-erbα* expression in the cornea, and decreasing corneal sensitivity. This is consistent with published evidence of short-term HFD-induced disruption of molecular circadian rhythms in mice [80]. The potential significance of the diminished influx of neutrophils may be revealed in similar findings by other investigators analyzing circadian changes in circulating neutrophils and their migration into diverse tissues [81–83]. These studies in mice document an

"aging" process in the blood neutrophils after they leave the bone marrow, characterized by phenotypic changes evident in surface proteins such as CD11b, the α subunit of the adhesion molecule Mac-1, CD62L (the adhesion molecule L-selectin), and CXCR4, a chemokine receptor for CXCL12. CD11b and CXCR4 increase and CD62L decreases as neutrophils circulate [83, 84]. The number of neutrophils exhibiting these changes increases between noon and midnight each day with clearing from the blood by noon the following day, coincident with a peak in the daily increase in ICAM-1 (CD54) expression, an adhesive ligand on venules for the CD18 integrins [83, 85]. This relatively short time for neutrophils in the circulation has traditionally been considered evidence of a short life span if they are not recruited to a site of infection. However, recent work indicates that the fate of these cells is complex and important to innate and adaptive immunologic and homeostatic functions. Some neutrophils return to the bone marrow in response to the chemokine CXCL12, where they influence the production and release of new neutrophils. Others migrate into the spleen and lymph nodes where they play supporting roles for B and T cells, while some migrate into the liver influencing metabolic functions [86]. Thus, the observation that neutrophils exhibit a circadian flux within the limbal stroma of the cornea is consistent with the studies in other tissues, and the timing of the influx is also consistent with other tissues in the mouse.

The function of the circadian flux of neutrophils in the normal cornea is unknown, but our previous corneal wound studies reveal some possibilities. Neutrophils contain growth factors such as VEGF that are critical to nerve regeneration and epithelial cell division in the injured cornea [54]. A loss of the circadian flux of neutrophils at the limbus caused by consumption of a HFD for 10 days may be detrimental to corneal homeostasis, as there would be less VEGF available to contribute to nerve maintenance and epithelial cell division. Alternatively, the changes we observed might be dependent on a larger, overarching systemic change following HFD consumption. In ND mice, *Rev-erbα* expression in the cornea peaks at ZT 0–8 (when neutrophil accumulation in the limbus is lowest) and reaches its lowest expression level at ZT 12–20, when neutrophil accumulation is peaking in the corneal limbus. The synchronicity we observed between *Rev-erbα* and neutrophils is ablated in the HFD mice. This change in *Rev-erbα* is consistent with a study by Kohsaka et al., showing expression of circadian genes *Clock* and *Baml1* in adipose tissue is suppressed in mice on a short-term HFD [80].

Systemic inflammation alters wound healing in humans [87, 88]. Similarly, corneal inflammation caused by short-term HFD feeding alters corneal wound healing as evidenced by a reduction in limbal venule expansion, γδ T cell recruitment and platelet extravasation, as well as delayed epithelial wound closure and decreased basal epithelial cell division. By 48h post-injury, very few neutrophils migrated to the wound center and the neutrophils continued to increase at the limbus. γδ T cells are crucial to neutrophil recruitment [60] and their reduced numbers likely contributed to the inability of neutrophils to migrate to the central cornea. Moreover, the observed increase in limbal NK cells would tend to limit neutrophil migration [60]. Delayed wound healing was also evident at 96h post-injury with decreased epithelial thickness, decreased corneal sensitivity and delayed nerve recovery compared to mice fed a ND. Hence, it is clear that even a short-term 10 day HFD feeding alters the cornea to a level that impairs wound healing.

In summary, consumption of a HFD can cause pathological changes in corneal morphology and function, including disruptions to the circadian rhythm and an increased inflammatory state. While DiR corneas show improved nerve density and sensitivity compared to HFD corneas, elevated cytokine and chemokine levels persist in the cultured DiR corneas and their supernatants. These levels suggest that DiR, as employed in this study, does not fully restore the cornea to a baseline state. Overall, we show HFD consumption negatively impacts corneal homeostasis and diminishes the ability of the cornea to respond to injury, long before the

onset of hyperglycemia. Switching from a HFD to a ND only partially restores corneal homeostasis, suggesting a HFD may have a lasting negative impact on corneal health that may prove resistant to dietary therapeutic modulation.

## Supporting information

**S1 Rawdata. Contains raw data for Figs 1–7, Tables 1 and 2, and data mentioned within the text.** Values depicted graphically are bolded in each section.
(XLSX)

## Acknowledgments

The authors acknowledge the excellent technical assistance provided by Madhavi Chintalapati, Tiffany Bullock, Margaret Gondo and Zhijie Li.

## Author Contributions

**Conceptualization:** Aubrey Hargrave, Paul Landry, C. Wayne Smith, Alan R. Burns.

**Data curation:** Aubrey Hargrave, Justin A. Courson, Vanna Pham, Sam Hanlon.

**Formal analysis:** Aubrey Hargrave, Sri Magadi, C. Wayne Smith, Alan R. Burns.

**Funding acquisition:** Rolando E. Rumbaut, C. Wayne Smith, Alan R. Burns.

**Investigation:** Aubrey Hargrave, Vanna Pham, Sri Magadi, Pooja Shankar, Sam Hanlon, Apoorva Das.

**Methodology:** Aubrey Hargrave, C. Wayne Smith, Alan R. Burns.

**Project administration:** C. Wayne Smith, Alan R. Burns.

**Resources:** Rolando E. Rumbaut, C. Wayne Smith, Alan R. Burns.

**Software:** Justin A. Courson, Paul Landry.

**Supervision:** Rolando E. Rumbaut, C. Wayne Smith, Alan R. Burns.

**Validation:** Aubrey Hargrave, Justin A. Courson.

**Visualization:** Aubrey Hargrave.

**Writing – original draft:** Aubrey Hargrave, C. Wayne Smith, Alan R. Burns.

**Writing – review & editing:** Aubrey Hargrave, Justin A. Courson, Alan R. Burns.

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
