## [Decision Letter · Decision Letter 0]

14 Aug 2020

PONE-D-20-21463

­Corneal dysfunction precedes the onset of hyperglycemia in a mouse model of diet-induced obesity

PLOS ONE

Dear Dr. Hargrave,

Thank you for submitting your manuscript to PLOS ONE. After careful consideration, we feel that it has merit but does not fully meet PLOS ONE’s publication criteria as it currently stands. Therefore, we invite you to submit a revised version of the manuscript that addresses the points raised during the review process.

The paper has been reviewed quite favorably. Please revise it along the lines of the critique paying particular attention to the following points:

1. Please present data on systemic changes as a Table (maybe as part of Table 1) and the data for 1 day and 3 days on HFD with quantitation.

2. The raw data for IL-12p40 and for FACS of leucocyte subsets (presented in the text, need to be incorporated into the Supplemental Section.

3. Please consider changing the abbreviation “DR” for diet reversal to something else (like DiR) as it may be confused, especially in this context, with “diabetic retinopathy”.

We look forward to receiving your revised manuscript.

Kind regards,

Alexander V. Ljubimov, Ph.D.

Academic Editor

PLOS ONE

Journal Requirements:

2.Thank you for your ethics statement:

'Animals were treated according to the guidelines described in the Association for

Research in Vision and Ophthalmology Statement for Use of Animals in Vision and

Ophthalmic Research, Baylor College of Medicine Animal Care and Use Committee

policy (IACUC approval #AN2721), and Animal Care and Use Policies of the University

of Houston (IACUC approval #16-005). Animals were anesthetized by an

intraperitoneal injection of pentobarbital sodium solution (40 mg/kg body weight;

Nembutal, Ovation Pharmaceuticals, Deerfield, IL). Animals were euthanized by either

CO2 inhalation or isoflurane overdose, followed by subsequent cervical dislocation

according to IACUC guidelines.'

Please amend your current ethics statement to confirm that your named ethics committee specifically approved this study.

For additional information about PLOS ONE submissions requirements for ethics oversight of animal work, please refer to http://journals.plos.org/plosone/s/submission-guidelines#loc-animal-research  

Reviewers' comments:

Reviewer's Responses to Questions

**Comments to the Author**

1. Is the manuscript technically sound, and do the data support the conclusions?

Reviewer #1: Yes

2. Has the statistical analysis been performed appropriately and rigorously? 

Reviewer #1: Yes

3. Have the authors made all data underlying the findings in their manuscript fully available?

Reviewer #1: Yes

4. Is the manuscript presented in an intelligible fashion and written in standard English?

Reviewer #1: Yes

5. Review Comments to the Author

Reviewer #1: This paper presents a large amount of interesting and solid data showing that mice fed on high-fat diet (HFD) display corneal alterations preceding the start of persistent hyperglycemia and that diet reversal incompletely improved this dysfunction, which suggests HFD may have a long-term negative effect on corneal health.

The few comments I have are the following: the data on systemic changes in body composition, metabolism and leucocytes were presented only in the text (lines 189-201), whereas the data for 1 day and 3 days on HFD were merely descriptive (lines 190-193) lacking any quantification. They were also absent from the Supplemental Section that contains raw data for Figs 1-7. Similarly, the raw data for IL-12p40 (line 247) and for flow cytometric evaluation of leucocyte subsets (presented in the text, lines 267-270) are absent from Supplemental Section.

6. PLOS authors have the option to publish the peer review history of their article (what does this mean?). If published, this will include your full peer review and any attached files.

Reviewer #1: No

---

## [Author Response · Author response to Decision Letter 0]

19 Aug 2020

Re: PONE-D-20-21463, Corneal dysfunction precedes the onset of hyperglycemia in a mouse model of diet-induced obesity

Dear Editor, 

Thank you for the review and the opportunity to address the questions and comments. Please find enclosed our responses below. The comments are in bold and our responses are in plain text. Changes to the manuscript are indicated under track changes. Changes to the excel spreadsheets are as indicated.

We hope the changes to our manuscript are acceptable and that the manuscript is now suitable for publication.

Sincerely,

Aubrey Hargrave and Alan Burns

Response to Reviewer and Editor Comments:

1. “Please present data on systemic changes as a Table (maybe as part of Table 1) and the data for 1 day and 3 days on HFD with quantitation.”

“the data on systemic changes in body composition, metabolism and leucocytes were presented only in the text (lines 189-201), whereas the data for 1 day and 3 days on HFD were merely descriptive (lines 190-193) lacking any quantification. They were also absent from the Supplemental Section that contains raw data for Figs 1-7.”

Both Reviewer #1 and the Editor wanted to see the systemic data for the 10 week mice (described in lines 189-201) reported in tabular form. We have added a new Table 1 (line 203-204) for 10 week HFD systemic changes and updated the text accordingly. We renamed the previous table with 10 day HFD changes to Table 2 and updated the text to reflect this.

We apologize for the confusion in the Supplemental data – the changes in body composition (eAT) and flow cytometry data for the 10 week mice were located in the “Results Text” tab of the excel spreadsheet, along with all other data discussed in the text without a corresponding figure. In order to make these data easier to locate, we have placed it under a separate excel tab (Table 1-10w systemic data). 

Thank you for pointing out the discrepancy with the 1 day and 3 day HFD data (line 190-193). We were referring to data from our previously published study and neglected to end the sentence with a citation. The citation has now been added (#57).

2. “The raw data for IL-12p40 and for FACS of leucocyte subsets (presented in the text, need to be incorporated into the Supplemental Section.”

“Similarly, the raw data for IL-12p40 (line 247) and for flow cytometric evaluation of leucocyte subsets (presented in the text, lines 267-270) are absent from Supplemental Section.”

We again apologize for the confusion in the Supplemental data. As noted, raw data for the 10 week cytokines evaluated from whole corneas from line 247 (IL-12p40, as well as IL-1A and IL-10) and the flow cytometry data for the 10 day HFD mice (line 267-270) were previously located together in one “Results Text” tab of the excel spreadsheet. In order to make these data easier to locate, we have placed it into separate excel tabs (Text-10w whole cornea analytes and Text-10d systemic data). 

3. Please consider changing the abbreviation “DR” for diet reversal to something else (like DiR) as it may be confused, especially in this context, with “diabetic retinopathy”. 

We agree with the reviewer and editor, and have made the change from DR to DiR throughout the manuscript, as well as in Fig 2 and Fig 3, to avoid any confusion. 

We have reviewed the PLOS ONE style templates and believe the manuscript meets the style requirements. 

5. Please amend your current ethics statement to confirm that your named ethics committee specifically approved this study.

We amended the ethics statement as follows: 

“The ethics committees at Baylor College of Medicine Animal Care and Use (IACUC approval #AN2721), and at the University of Houston Animal Care and Use (IACUC approval #16-005) approved this study.”

This sentence was also updated within the Materials and Methods section to read:

“The Animal Care and Use ethics committees at Baylor College of Medicine (IACUC #AN2721), and at the University of Houston (IACUC #16-005) approved this study.”

---

## [Decision Letter · Decision Letter 1]

24 Aug 2020

Corneal dysfunction precedes the onset of hyperglycemia in a mouse model of diet-induced obesity

PONE-D-20-21463R1

Dear Dr. Hargrave,

We’re pleased to inform you that your manuscript has been judged scientifically suitable for publication and will be formally accepted for publication once it meets all outstanding technical requirements.

Kind regards,

Alexander V. Ljubimov, Ph.D.

Academic Editor

PLOS ONE

Additional Editor Comments (optional):

Reviewers' comments:

Reviewer's Responses to Questions

**Comments to the Author**

1. If the authors have adequately addressed your comments raised in a previous round of review and you feel that this manuscript is now acceptable for publication, you may indicate that here to bypass the “Comments to the Author” section, enter your conflict of interest statement in the “Confidential to Editor” section, and submit your "Accept" recommendation.

Reviewer #1: All comments have been addressed

2. Is the manuscript technically sound, and do the data support the conclusions?

Reviewer #1: Yes

3. Has the statistical analysis been performed appropriately and rigorously? 

Reviewer #1: Yes

4. Have the authors made all data underlying the findings in their manuscript fully available?

Reviewer #1: Yes

5. Is the manuscript presented in an intelligible fashion and written in standard English?

Reviewer #1: Yes

6. Review Comments to the Author

Reviewer #1: (No Response)

7. PLOS authors have the option to publish the peer review history of their article (what does this mean?). If published, this will include your full peer review and any attached files.

Reviewer #1: No

---

## [Editor Report · Acceptance letter]

28 Aug 2020

PONE-D-20-21463R1 

­Corneal dysfunction precedes the onset of hyperglycemia in a mouse model of diet-induced obesity 

Dear Dr. Hargrave:

I'm pleased to inform you that your manuscript has been deemed suitable for publication in PLOS ONE. Congratulations! Your manuscript is now with our production department. 

Kind regards, 

on behalf of

Dr. Alexander V. Ljubimov 

Academic Editor

PLOS ONE